# Cognitive Load Moderates the Effects of Total Sleep Deprivation on Working Memory: Evidence from Event-Related Potentials

**DOI:** 10.3390/brainsci13060898

**Published:** 2023-06-01

**Authors:** Ying Yin, Shufang Chen, Tao Song, Qianxiang Zhou, Yongcong Shao

**Affiliations:** 1Key Laboratory for Biomechanics and Mechanobiology of the Ministry of Education, School of Biological Science and Medical Engineering, Beihang University, Beijing 100191, China; yingyin_1234@163.com (Y.Y.); budeshao@aliyun.com (Y.S.); 2School of Psychology, Beijing Sport University, Beijing 100084, China; chenshufang0617@126.com (S.C.); songtaozy@163.com (T.S.)

**Keywords:** sleep deprivation, working memory, cognitive load, event-related potentials, electroencephalography, N-back

## Abstract

**Purpose**: The function of working memory (WM) is impaired by total sleep deprivation (TSD) and cognitive load. However, it is unclear whether the load modulates the effect of TSD on WM. We conducted a pilot study to investigate the effects of 36 h of TSD on WM under different load levels. **Materials and methods:** Twenty-two male students aged 18–25 years were enrolled, who underwent two types of sleep conditions (baseline and 36 h TSD), where they performed two N-back WM tasks (one-back task and two-back task) with simultaneous electroencephalography recordings. **Results:** Repeated measures analysis of variance (ANOVA) indicated that, with the increasing load, the reaction time increased and the accuracy decreased. After TSD, the correct number per unit time decreased. The significant interaction effect of the P3 amplitudes between the load level and the sleep condition showed that the reduction in the amplitude of P3 in the two-back task due to TSD was more obvious than that in the one-back task. **Conclusions:** Our results provided evidence for the moderation of load on the impairment of TSD on WM. The degree of TSD-induced impairment for a higher load was greater than that for a lower load. The current study provides new insights into the mechanisms by which sleep deprivation affects cognitive function.

## 1. Introduction

With the continuous development of a social economy, quality of life is improving, and the quality of sleep has also received widespread attention. However, the overall sleep status of people is poor. Many people spend long hours in work and education and are often unable to get enough sleep. Sleep loss continues to cause stress for individuals. Therefore, several researchers have used the method of total sleep deprivation (TSD) to explore the impact of inadequate sleep on individuals. TSD affects cognitive function [1], such as working memory (WM). WM is a memory system with a limited capacity for temporary storage and processing of information, which plays an important role in various cognitive tasks [2]. Patients with sleep disorders have a weak WM performance because of the reduced activation and functional connectivity in the prefrontal subregions [3]. TSD can disrupt many aspects of WM [4]. For example, a study found that TSD impaired the objective but not the self-estimated performance of WM in women [5]. Some studies have also shown that TSD can impair spatial WM [6,7]. In addition, verbal WM is also impaired by TSD [8].

There are several studies regarding brain mechanisms involved in the effects of sleep deprivation on WM. In a review, the authors searched for empirical literature on the effects of TSD or sleep restriction on WM and concluded that WM performance decreases after TSD, which was associated with changes in neural activation in the brain [4]. A functional magnetic resonance imaging study has shown that TSD may affect WM by altering the functional connectivity between the default mode network, the dorsal attention network, and the frontoparietal network [9]. A neuroimaging study found reduced signals in extra-striate (visual) cortical areas during the execution of WM tasks after TSD [1]. A near-infrared spectroscopy study showed insufficient frontal activation in WM processing after TSD [10]. Several electroencephalography (EEG) studies have also provided evidence for the negative effects of TSD on WM. A study explored the electrophysiological mechanisms of WM alteration after TSD and showed that information replacement in WM was reduced after 36 h of TSD [11]. Two studies have demonstrated that brain regions associated with memory (e.g., hippocampus) were impaired by TSD [12], which also led to dysfunctional consolidation and reconsolidation during memorization [13]. In conclusion, a large number of studies showed that TSD has a negative impact on cognitive function.

Research related to WM shows that cognitive load is one of the important factors affecting WM. Cognitive load is the total amount of cognitive resources that a person needs to expend to process information while completing a task or learning. A study found that a high cognitive load reduced the accuracy of color and shape recall, suggesting that a high load impaired WM performance [14]. Some researchers found reduced performance on the visual WM under a high load level that required the memorization of seven random digits [15]. Using load manipulation, a study found that cognitive load could influence WM performance by limiting cognitive resources in WM [16]. As we know, WM performance will be impaired after TSD. Engineering psychology suggests that individuals have large variations in load during fatigue, and studying load may help cope with the decline in WM caused by TSD. N-back tasks using different types of stimuli (English letters, small black squares, and geometric figures) have been studied to examine differences in the effects of TSD on different WM, including pronunciation WM, spatial WM, and object WM [7]. However, the role of load, one of the important factors of WM, in the relationship between TSD and WM and its neurophysiological mechanisms is not yet known. A functional magnetic resonance imaging (fMRI) study explored the brain’s response to increased WM load following TSD, finding that WM was impaired [12]. As the temporal resolution of EEG is better than that of fMRI, it is more suitable for studying electrophysiological mechanisms. Thus, we intend to address this issue using EEG technology in our study.

In the large body of existing experimental research on WM, the frequently used methods of cognitive manipulation include Sternberg-type verbal WM tasks [17,18] and the N-back task [19]. The N-back task is based on the inverse of the N-test paradigm. The N-back task can successfully predict individual differences in higher cognitive functions, such as fluid intelligence, in tasks with higher load levels [20]. N-back tasks have been used in several studies related to WM [21,22,23]. Two studies reported alpha and beta modulation over anterior motor regions, which suggested an influence of WM during the encoding and maintenance stages of an N-back task [24,25]. It may be better to distinguish only between one-back and two-back load studies, which reflect increases in retention and the need for updating [26]. Therefore, the present study aims to explore the variety of WM performance before and after TSD using pronunciation N-back tasks with different load levels.

We also explored the time course of the above problem, and EEG helped us understand the neural basis of WM. Event-related potentials (ERPs) are used to explore this question, owing to their high temporal resolution, which can help us understand the neural basis of WM. Several ERP studies have used N2 and P3 waves to reflect potential physiological indicators of TSD effects [11,27,28,29,30]. Previous models of perception often claim that bottom-up processing occurs in an early time window, while top-down control occurs in a late time window after stimulus onset [31]. The N2 component is a negative deflection with a latency between 200 and 400 ms and is often an important indicator of conflict monitoring [11,32,33,34]. N2 modulations are more readily attributed to the orientation of visual attention and detection of novel stimuli [35], which belong to the part of bottom-up processing. The P3 component is a late visually evoked ERP, which is usually shown within a 250–500 ms time window after stimulus presentation; its maximum positive deflection is among the parietal (P3, Pz, P4) electrodes [36]. The top-down control reflected by the P3 component includes the judgment of stimulus consistency in WM, the process of decision-making, the process of memory updating [37], and the deployment of attention resources [38,39]. The P3 component revealed significant effects of top-down attention [40]. A study separated bottom-up processing and top-down control, and considered that the P3 component reflected top-down processing and the N2 component reflected bottom-up processing [27]. The study showed a decrease in P3 amplitude after TSD, which indicated that top-down control was impaired, but bottom-up processing was not impaired. Therefore, inspired by previous research, this study used the components N2 and P3 to quantify the bottom-up processing and the top-down control, respectively.

The aim of this study was to verify the effect of TSD on pronunciation WM and to explore the load effect between them. Therefore, we conducted 36 h TSD experiments and measured N-back task pronunciation with different load levels. As the individual differences in WM efficiency had a great influence on the experiment, high WM was beneficial to resist WM load and inhibit irrelevant information. Accordingly, we adopted a within-subject, transverse, and interventional design. Reaction time (RT) and accuracy were measured as behavioral indicators. We also recorded EEG data to assess the changes in the P3 and N2 components. We assumed the following: (a) TSD can impair the functions of WM and (b) load will moderate the effects of TSD on WM.

## 2. Materials and Methods

### 2.1. Participants

Twenty-two healthy right-handed male students were recruited by a campus advertisement as participants in this study. The participants were included if (1) they were between 18 and 25 years old; (2) they had normal or corrected normal vision; (3) their intelligence scores was >110 on the Raven test; (4) they did not smoke or drink alcohol for at least one month before the start of the experiment; (5) they had no hair coloring, perming, or bleaching within three months; (6) they had no physical or mental illness; and (7) they had good sleep habits (Pittsburgh Sleep Quality Index, PSQI < 5). PSQI is one of the most validated and widely used sleep disorder assessment scales. The scale consists of nine questions; the first four are fill-in-the-blank and the last five are multiple-choice questions. The total score ranges from 0 to 21 points and, the higher the score, the worse the sleep quality. Each participant signed written informed consent. The experiment was approved by the Ethics Committee of Beihang University.

### 2.2. Experimental Design

The N-back tasks require the participants to compare the stimulus that has just appeared with the nth stimulus in front of it, and the load is manipulated by controlling the number of stimuli between the current stimulus and the target stimulus. When *n* = 1, participants are asked to compare the current stimulus to the previous stimulus adjacent to it. When *n* = 2, the current stimulus is compared to the stimulus located one position in front of it. This study used the one-back and two-back pronunciation work memory tasks (Figure 1). The stimuli for this task were 14 uppercase and lowercase English letters. Participants used the pronunciation of letters to determine whether two stimuli were consistent. Then, they were asked to pay attention to the pronunciation of each letter while they ignored their case. The letters with similar cases were excluded, such as W/w, and Z/z. The specifics of the tasks have been provided previously [7]. All materials, shown in black, were presented on a white background with an approximate visual angle of subtending of 1.5° × 1.5° (height: 2.0 cm, width: 2.0 cm). There were 122 trials for each task. The black ‘+’ appeared as stimulus onset asynchrony time for 1600 ms, then the target stimulus appeared for 400 ms. In each one-back task trial, the current stimulus appeared after the first target stimulus was presented. Further, in each trial of the two-back task, the first target stimulus was presented and the current stimulus appeared at intervals. When the two were identical (match), each participant was required to click the left mouse button, while clicking the right mouse button when they were inconsistent (mismatch). The match and mismatch conditions were presented with a 1:1 ratio in a pseudorandom order.

### 2.3. Experimental Procedures

The experimental procedure is illustrated in Figure 2. Before the experiment, all participants were informed about the WM tasks and practiced the one-back and two-back pronunciation tasks. Two participants simultaneously conducted the experiment with a fixed order of N-back tasks. The day before the experiment, participants were invited to the laboratory and slept there at night. The first WM tasks were performed from 7:30 a.m. to 8:30 a.m. the next morning, with simultaneous EEG recording (baseline). After 36 h of TSD, they completed similar pronunciation WM tasks, with the second simultaneous EEG recording (TSD) from 7:30 p.m. to 8:30 p.m. on the third day. For one participant, the order of the two tasks was fixed under two sleep conditions. Between the two participants, the order of the two N-back tasks was counterbalanced. Participants could watch movies, play computer games, read books, chat, and so on, but they did not drink alcohol, tea, or coffee, or smoke or take any medication during the TSD experiment. Furthermore, they were accompanied, observed, and reminded by the nursing staff.

Before and after the 36 h TSD, we measured an indicator to verify the manipulation of the TSD. The indicator was the sleepiness/wakefulness scale extracted from the visual analog scale [5,11], which reflected subjective sleepiness. This scale includes the wakefulness scale and the sleepiness scale. These two subscales were divided into 10 levels, from 0 (low) to 9 (high). A higher score on the sleepiness subscale and a lower score on the wakefulness subscale was used to assess the success of our TSD manipulation. All participants were required to complete these parts before performing N-back pronunciation tasks.

### 2.4. Data Analysis of Behavioral Experiments

Behavioral data, including reaction time (RT), accuracy, and the correct number per unit time, were recorded for analysis in the different loads of pronunciation N-back tasks under both baseline and 36 h TSD conditions. The analyses were processed by IBM SPSS (Version 22.0), in which repeated measures analysis of variance (ANOVA) with Bonferroni post hoc analysis was performed. Descriptive data are presented as the mean ± standard deviation (SD). The main effects and interaction between sleep conditions (baseline and TSD) and load levels (one-back and two-back) were analyzed.

### 2.5. EEG Recordings and Preprocessing

While participants performed the pronunciation WM task, continuous EEG was recorded using a BrainVision recorder amplifier (Brain Products). An elastic cap with 30 electrodes (Easycap, BrainProducts GmbH, Gilching, Germany) was mounted according to the standard positions of the 10–20 system. The online EEG sampling rate was 1000 Hz. The conductive gel was applied on the recording sites using blunt swabs, while an abrasive cream and alcohol wipes were used to clean the mastoids and the forehead before electrodes were attached. The impedance of all channels was reduced and maintained below 10 kΩ. Electrodes were placed above and below the left eye to record the vertical eye movement and on the bilateral temples to monitor the horizontal eye movement. The online reference electrodes were placed on the bilateral mastoids and the forehead was used as a base.

The EEGLAB2020_0 toolbox of MATLAB R2017a [41] was used to pre-analyze the raw EEG data. A bandpass filter from 0.1 to 30 Hz with a frequency slope of 24 dB/oct was used through a fourth-order Butterworth filter in the ERPLAB plugin [42]. Re-reference was the average reference and the sampling rate was reduced to 250 Hz. After independent component analysis, components representing eye movement and excessive muscle activity were identified by the ICLABEL plugin [43] and were manually removed. In the one-back task, the mean RT from baseline was 569.36 ± 72.15 ms and the mean RT from TSD was 617.38 ± 107.93 ms. In the two-back task, the mean RT from baseline was 731.37 ± 143.45 ms and the mean RT from TSD was 733.99 ± 176.86 ms. Hence, epochs with a 1000 ms length ranging from −200 ms to 800 ms were extracted from the continuous EEG data. Then, a baseline correction was performed by subtracting a pre-stimulus period of the mean amplitude from −200 ms to 0 ms. The trials with a voltage exceeding ±75 μV were removed from the gross average ERPs. Data from two participants were excluded from the statistical analysis because their signal-to-noise ratios were too low. Thus, the final sample consisted of 20 participants.

### 2.6. ERP Data Analysis

After averaging and calculating according to the corrected responses, the characteristics of the ERP components were extracted using the ERPLAB plugin [42]. Two components, N2 (200–350 ms) and P3 (300–800 ms), were identified and quantified according to the waveforms. The mean gross amplitudes and fractional area latency of the N2 and P3 components were calculated at the Fz and Pz electrodes. A 2 (sleep conditions: baseline, TSD) × 2 (load levels: one-back, two-back) repeated measures ANOVA with Bonferroni post hoc tests was used to analyze the components N2 and P3.

## 3. Results

### 3.1. Manipulation Checks

Statistical analysis of the paired sample *t*-test showed that the different scores of the sleepiness subscale at baseline (2.18 ± 1.50) and after TSD (6.58 ± 1.83) were significant (*t*_21_ = 11.19, *p* < 0.001, Cohen’s d = −1.58). The different scores on the wakefulness subscale at baseline (8.05 ± 1.21) and after TSD (4.68 ± 1.69) were significant (*t*_21_ = −7.41, *p* < 0.001, Cohen’s d = −1.58). The results showed that all participants felt more sleepy and less awake after TSD, which indicated that the manipulation of TSD was effective.

### 3.2. Behavioral Performance

Characteristics of behavioral performance are shown in Table 1.

Repeated measures ANOVA of RT revealed a significant main effect of load levels [*F*_(1,21)_ = 39.06, *p* < 0.001, partial η^2^ = 0.65], suggesting that RT was longer at the two-back task than that at the one-back task. The interaction effect of the load level with the sleep condition was not significant [*F*_(1,21)_ = 2.21, *p* = 0.152, partial η^2^ = 0.10]. The RT of the main effect of the sleep condition was not significant [*F*_(1,21)_ = 1.08, *p* = 0.310, partial η^2^ = 0.05].

ANOVA revealed that the main effect of the accuracy load level was significant, showing the differences between the one-back task and the two-back task [*F*_(1,21)_ = 16.69, *p* = 0.001, partial η^2^ = 0.44]. There was no significant interaction effect between sleep conditions and load levels [*F*_(1,21)_ = 0.77, *p* = 0.389, partial η^2^ = 0.04]. The main effect of sleep condition was not significant [*F*_(1,21)_ = 2.51, *p* = 0.128, partial η^2^ = 0.11].

Repeated measures ANOVA of the correct number per unit time revealed a significant interaction between sleep condition and load level [*F*_(1,21)_ = 4.74, *p* = 0.041, partial η^2^ = 0.18]. Further simple effects analysis showed that the correct number per unit time decreased after TSD in the one-back task, while it increased slightly after TSD in the two-back task. The main effect of sleep condition was significant [*F*_(1,21)_ = 39.08, *p* < 0.001, partial η^2^ = 0.65]. The main effect of load level was not significant [*F*_(1,21)_ = 1.14, *p* = 0.299, partial η^2^ = 0.05].

### 3.3. ERP Results

#### 3.3.1. N2 Component

Repeated measures ANOVA of mean amplitudes (Table 2 and Figure 3) of the N2 component showed that the interaction effect of load levels in sleep conditions was not significant [*F*_(1,19)_ = 0.47, *p* = 0.502, partial η^2^ = 0.02]. The main effect of sleep conditions [*F*_(1,19)_ = 0.92, *p* = 0.350, partial η^2^ = 0.05] and the main effect of load levels [*F*_(1,19)_ = 0.22, *p* = 0.648, partial η^2^ = 0.01] were not significant. As can be seen from the topographic map of N2 (Figure 3), the increased load in the TSD condition caused the color bar to fade. This partly reflected TSD and load effects, although there was no statistically significant difference.

Repeated measures ANOVA of N2 latency (Table 2) revealed that the interaction effect between sleep conditions and load levels was not significant [*F*_(1,19)_ = 0.79, *p* = 0.386, partial η^2^ = 0.04]. Furthermore, the main effect of sleep conditions [*F*_(1,19)_ = 0.12, *p* = 0.732, partial η^2^ = 0.006] and the main effect of load levels [*F*_(1,19)_ = 0.44, *p* = 0.516, partial η^2^ = 0.02] were not significant.

#### 3.3.2. P3 Component

Repeated measures ANOVA of the mean amplitude (Table 3 and Figure 4) of the P3 component revealed that the interaction effect of sleep condition by load level was significant [*F*_(1,19)_ = 5.33, *p* = 0.032, partial η^2^ = 0.22]. Further simple effects analysis showed that, in the one-back task, the difference in amplitudes between different sleep conditions was not large, while in the two-back task, the P3 amplitude due to TSD was more negative than that in the baseline. A significant main effect of load levels [*F*_(1,19)_ = 10.72, *p* = 0.004, partial η^2^ = 0.36] suggested that the mean amplitudes triggered by the two-back task were lower than those triggered by the one-back task. The main effect of the sleep condition was not significant [*F*_(1,19)_ = 0.70, *p* = 0.415, partial η^2^ = 0.04]. The topographic map of P3 (Figure 4) showed a large change in the color bar in the central parietal region. This also proved that TSD and load had an impact on WM performance.

Repeated measures ANOVA of the latencies (Table 3) of the P3 component revealed that the main effect of sleep condition was significant [*F*_(1,19)_ = 4.77, *p* = 0.042, partial η^2^ = 0.20], suggesting that the latencies of the P3 waves under the TSD condition were longer than those at baseline. A significant main effect of load levels [*F*_(1,19)_ = 6.06, *p* = 0.024, partial η^2^ = 0.24] suggests that the latencies triggered by the two-back task were longer than those triggered by the one-back task. The interaction effect between sleep condition and load level was not significant [*F*_(1,19)_ = 0.01, *p* = 0.909, partial η^2^ = 0.001].

## 4. Discussion

In this study, we performed a 36 h TSD experiment and two N-back WM tasks that represented different loads with simultaneous EEG recordings. We analyzed behavioral data and ERP characteristics between two sleep conditions (baseline and TSD) and two load levels (one-back and two-back). The main effect of sleep conditions and the interaction of the two factors of the correct number per unit time were significant. Moreover, the interaction of the two factors of P3 was also significant. The results showed that TSD impaired WM, and this effect was moderated by the cognitive load.

The behavioral results showed that the correct number per unit time decreased after TSD, meaning that WM was impaired by TSD. A previous study confirmed this result [7]. The correct number per unit time decreased after TSD in one-back tasks, while it increased slightly after TSD in two-back tasks. This suggested that the load may have modulated the TSD effect. The load manipulation in this study was successful owing to the increased RT and decreased accuracy when the load of the pronunciation N-back task increased, which was consistent with the conclusions of previous studies [16,44]. RT tended to be longer and accuracy tended to be lower after TSD than those at the baseline; however, the differences were not significant. Similar results were found in the N2 component. The changes in amplitude and latency in the N2 component were not significant. A possible reason was that TSD effects canceled out the effects of load. Participants needed more cognitive resources to complete the two-back task because the load of the two-back task was larger than that of the one-back task. Furthermore, the activation of ERP components may increase; however, after TSD, participants’ brain region activation may be reduced. Therefore, the two effects may cancel each other out, resulting in insignificant changes.

The most important finding of the current study was the significant interaction effect of load level and sleep condition on the P3 amplitudes. This result showed that the load moderates the effects of TSD on WM. At different load levels, the changed degree of the P3 amplitudes was obviously inconsistent across the two sleep states. In the higher load task, TSD resulted in obviously lower amplitudes of the P3 component compared with that at the baseline. In the lower load task, TSD resulted in a slightly higher amplitude of the P3 components compared with that at the baseline; however, it was not statistically significant. Increased task difficulty or complexity followed by TSD might lead to decreased activation with WM decline [45,46]. According to a previous study, TSD reduces individuals’ motivation to complete tasks [47]. In lower-load (one-back) tasks, individuals could still maintain performance after TSD because they believe that hard work will keep their performance stable or even up. However, in higher-load (two-back) tasks, individuals may feel inadequate, obviously reducing their motivation to maintain performance, resulting in weakened activation of brain regions. In other words, individuals who undergo TSD may allocate more compensatory effort to easier tasks and choose to detach from more challenging tasks [48]. The interaction effect on RT or accuracy was not significant, whereas it was significant on the P3 amplitude. This was mainly due to the ERP indicators with high time resolution being more sensitive than the behavioral performance indicators. The decrease in P3 amplitude showed that the processing of the matching response was altered to some extent after TSD [49], and cognitive resources were insufficient with increasing load [50].

This study also revealed a longer latency of the P3 component for 36 h TSD than that at the baseline, showing that brain activities were slowed down. TSD reduced cortical arousal and impaired vigilant attention [51,52]. Two studies attributed the impairment of WM due to TSD to the impairment of control of attentional resources and the speed of processing information for individuals [53,54]. There were also previous studies with similar results, suggesting that TSD led to a slowdown in brain activities, representing a breakdown in top-down cognitive control [7,28,55].

Combined with the results of two ERP components, we found that TSD caused a breakdown in the top-down cognitive control, rather than an impairment in the bottom-up automatic processing. Changes in N2 amplitude and latency were not obvious under different conditions, which meant that bottom-up processing had greater flexibility to counteract the effects of TSD and higher load. However, the ERP components of top-down control showed a high vulnerability to TSD. The amplitude and latency of the P3 component during different conditions had noticeable changes, revealing that the top-down control was altered by TSD and a higher load. More importantly, the interaction effect between sleep conditions and load levels of the P3 amplitudes was significant. This was evidence for the moderation effects of load on TSD and WM, and the moderation effects of load were reflected in the top-down control.

This study has some limitations. First, the sample was small and all participants were men, which may affect the generalizability of the results. A larger sample, including women, is warranted to confirm the results. Second, in terms of the project’s implementation, we did not consider the differences between other age groups, such as children and the elderly; therefore, the extrapolation of the results requires caution. Third, although sleep quality was measured at the time of recruitment, PSG was stricter and more helpful than PSQI. It is hoped that future studies will use PSG. Fourth, combining EEG recordings with other cognitive neuroscience tools would facilitate a clearer picture of the effects of TSD on WM and the role of load between them.

## 5. Conclusions

In conclusion, our research showed that TSD caused a breakdown in top-down cognitive control, whereas bottom-up automatic processing seemed more resilient against the detrimental effects of TSD. Moreover, the moderation of load on the effects of TSD on WM existed in the top-down cognitive control. Persistent sleep restriction impacts neurocognition in a manner similar to laboratory-controlled TSD. In a state of fatigue caused by lack of sleep, the individual’s load increases significantly. Our study provides electrophysiological evidence to understand the mechanism under the moderation of load between TSD and WM. Load studies can help improve work performance in response to chronic sleep deprivation. We hope to provide some help with sleep problems from the neuro-electrophysiological viewpoint.

## Figures and Tables

**Figure 1 brainsci-13-00898-f001:**
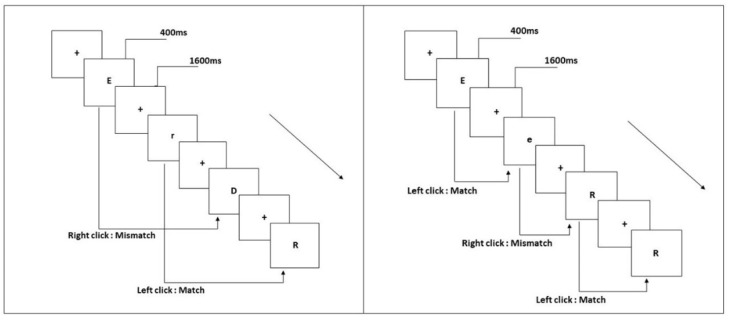
Pronunciation working memory tasks. (**Left**) two-back; (**right**) one-back.

**Figure 2 brainsci-13-00898-f002:**
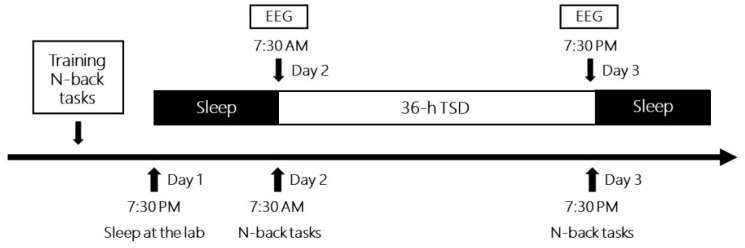
Timeline of total sleep deprivation experiments for each participant. EEG, electroencephalography.

**Figure 3 brainsci-13-00898-f003:**
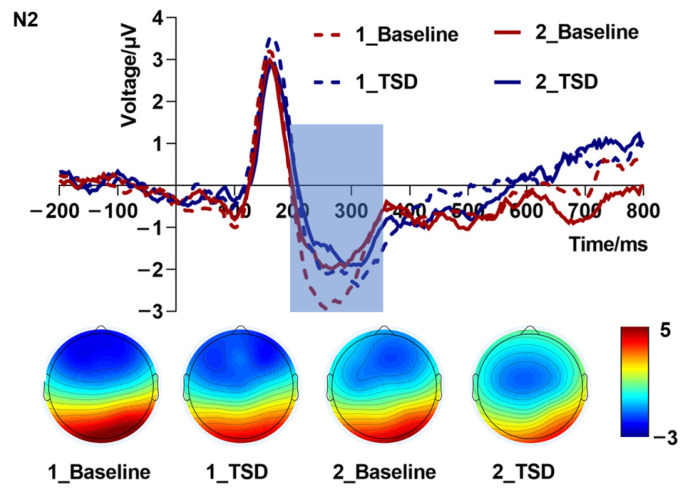
N2 amplitude (200–350 ms, Fz) at baseline and 36 h total sleep deprivation (TSD) for the correct response to N-back working memory tasks.

**Figure 4 brainsci-13-00898-f004:**
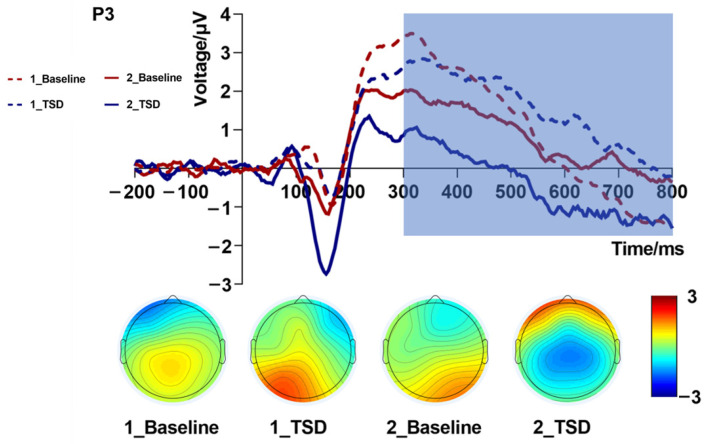
P3 amplitude (300–800 ms, Pz) at baseline and 36 h total sleep deprivation (TSD) for the correct response to N-back working memory tasks.

**Table 1 brainsci-13-00898-t001:** Summary of behavioral performance (n = 22, mean ± SD).

Behavior	Baseline		TSD	
	One-Back	Two-Back	One-Back	Two-Back
RT (ms)	569.36 ± 72.15	731.37 ± 143.45	617.38 ± 107.93	733.99 ± 176.86
Accuracy (%)	82.57 ± 15.73	73.79 ± 15.24	80.17 ± 14.93	67.88 ± 19.51
Correct number/s	1.78 ± 0.21	1.42 ± 0.29	1.66 ± 0.27	1.44 ± 0.35

RT, reaction time; TSD, total sleep deprivation.

**Table 2 brainsci-13-00898-t002:** The gross mean amplitude and latency of the N2 component (200–350 ms, Fz) in the correct response at baseline and in TSD (n = 20, mean ± SD).

Load	Baseline		TSD	
	Amplitude (μV)	Latency (ms)	Amplitude (μV)	Latency (ms)
One-back	−3.57 ± 3.33	268.00 ± 29.96	−2.75 ± 3.79	265.20 ± 33.85
Two-back	−3.13 ± 1.92	267.40 ± 40.23	−2.78 ± 3.09	275.80 ± 36.94

TSD, total sleep deprivation.

**Table 3 brainsci-13-00898-t003:** The gross mean amplitude and latency of the P3 component (300–800 ms, Pz) in the correct response at baseline and in TSD (n = 20, mean ± SD).

Load	Baseline		TSD	
	Amplitude (μV)	Latency (ms)	Amplitude (μV)	Latency (ms)
One-back	0.86 ± 2.28	507.20 ± 80.94	1.49 ± 2.43	547.20 ± 73.52
Two-back	0.63 ± 1.50	543.20 ± 89.98	−0.92 ± 3.12	586.40 ± 67.94

TSD, total sleep deprivation.

## Data Availability

The datasets generated for this study are available upon request from the corresponding author.

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
