# Peer review of "Cognitive Load Moderates the Effects of Total Sleep Deprivation on Working Memory: Evidence from Event-Related Potentials"

_brainsci, 2023, doi:10.3390/brainsci13060898_

Round 1

Reviewer 1 Report

Comments and Suggestions for Authors

The problem of sleep disorders is one of the most actual in modern society and clinical medicine. Sleep disorders have various adverse consequences for human health, among which cognitive disorders occupy a special place.

The article submitted for review is devoted to this complex problem and corresponds to the subject of the journal.

However, the manuscript needs revision.

Abstract: it is necessary to structure it (purpose, materials and methods, results, conclusion). What methods were used? What is the age of the participants? What are the comparison groups? What were the values of the P3 peak? Etc.

Introduction: delete all links older than 5 years, update the studied problem based on the results of research in recent years. Cognitive disorders, including working memory disorders induced by sleep disorders, are actively studied by various research groups. What's new? What remains unresolved? What prompted you to conduct this research?

Please indicate the purpose of your research at the end of the Introduction section.

Materials and Methods: Please clearly identify the inclusion and exclusion criteria, including the sex and age of the participants (is is well known, sex and age significantly affect the latency and amplitude of the P3 peak). Specify the units of measurement when using scales and questionnaires (points). Please specify the type of our study. Was it prolonged or transverse, open or closed, observational or interventional? What diagnostic equipment have you used to investigate cognitive evoked potentials?

It is necessary to revise the design of headings (please look at the template), links (remove surnames and years, add the link number in square brackets), tables (all columns in tables should have names).

The design of the References needs revision (please see the template and/or previously published articles in this journal).

Comments on the Quality of English Language

A minor improvement in the style of English is required.

Reviewer 2 Report

Comments and Suggestions for Authors

"Cognitive Load Moderates the Effects of Total Sleep Deprivation on Working Memory: Evidence from Event-Related Potentials" is an interesting study,

but

- the sample size is very small - it's only enough for a pilot study

- only PSQI is used as sleep evaluation - gold standard is PSG

- methods and results are too short - should be extended with more informations

- conclusion is missing

- literature has to be changed in sense of journal

- language has to be improved 

Comments on the Quality of English Language

Language has to be improved - e.g. grammar, orthography & sentences. It should be cheched by a native speaker.

Reviewer 3 Report

Comments and Suggestions for Authors

Thanks for recommending me as a reviewer. In this paper, twenty two male students were enrolled, who underwent two types of sleep conditions (baseline and 36 h-TSD), where they performed two N-back WM tasks (one-back task and two-back task) with simultaneous electroencephalography recordings. If the authors complete minor revisions, the quality of the study will be further improved.

1.The introduction section is well written. 

2. line 116-124: Authors should more clearly describe exclusions and inclusion criteria for their studies in the Methods section.

3. It is recommended to separate the conclusion section from the discussion section.

Round 2

Reviewer 1 Report

Comments and Suggestions for Authors

The authors modified the manuscript and its quality improved.

The article may be published after a minor revision:

1) it is necessary to add the name of the first column in all tables;

2) most of the headings still need to be arranged according to the paper template (for example, not an INTRODUCTION, but an Introduction; please also correct: DATA AVAILABILITY STATEMENT, MATERIALS AND METHODS, RESULTS, DISCUSSION, DATA AVAILABILITY STATEMENT, etc.); 

3) please add the numbers of sections and subsections to improve the reading of the article according to the paper template (examples: 1. Introduction; 2. Materials and methods; 2.1. Participants; 2.2. Experimental procedures, etc.).

Comments on the Quality of English Language

Minor editing of the English text is required.

In general, I did not have any difficulties with reading the article.
